# Improving Cryopreservation Efficiency and Pregnancy Rate through Superovulation with Follicle-Stimulating Hormone in Korean Hanwoo Cows via Ovum Pick Up

**DOI:** 10.3390/vetsci10090578

**Published:** 2023-09-18

**Authors:** Daehyun Kim, Junkoo Yi

**Affiliations:** 1Department of Animal Science, Chonnam National University, Gwangju 61186, Republic of Korea; chunja2411@naver.com; 2School of Animal Life Convergence Science, Hankyung National University, Anseong 17579, Republic of Korea; 3Gyeonggi Regional Research Center, Hankyong National University, Anseong 17579, Republic of Korea

**Keywords:** follicle-stimulating hormone, Hanwoo cow, ovum pick up, cryopreservation, embryo development, embryo transfer

## Abstract

**Simple Summary:**

Ovum pick up (OPU) methods play a significant role in oocyte collection worldwide and are the most efficient means of enhancing genetic advancement through maternal lines in cattle. The primary objective of this study was to establish an effective system for producing transferable embryos from high-quality cows using the OPU method. Oocytes were collected from Hanwoo cows, with one group administered follicle-stimulating hormone (FSH) and the untreated group serving as a control. The FSH group exhibited a higher proportion of grade A and B oocytes than those of the other grades, with most of the cumulus–oocyte complexes (COCs) at the germinal vesicle (GV) 2 stage. Additionally, the FSH group exhibited a significantly greater rate of blastocyst formation compared to the control group. The FSH group, after vitrification and in vitro culture, demonstrated superior re-expansion rates compared to the control group, showcasing the efficiency of FSH treatment in terms of embryo production, freezing, and preservation.

**Abstract:**

The aim of this study was to devise an efficient technique for generating embryos from high-quality bovine females. Oocytes were collected from 20 control and 15 Hanwoo (*Bos taurus coreanae*) females treated with the FSH. A combination of decreasing FSH doses (36, 36, 24, and 24 mg, 12 h apart), progesterone, estrogen, and prostaglandins were administered to synchronize and mildly stimulate the animals. The FSH-treated group (1125 oocytes) and control group (1022 oocytes) exhibited a higher proportion of grade A and B oocytes (88.2%) than the other grades (*p* < 0.05), with most at the germinal vesicle 2 stage (64.0%). Moreover, the FSH-treated group achieved a notably higher blastocyst rate (44.7%) compared to the control group (31.1%) (*p* < 0.05). After undergoing vitrification and in vitro culture (IVC) warming, embryos in the FSH group exhibited higher re-expansion rates (grade 1: 86.9%; grades 2 and 3: 57.9%) compared to those in the control (*p* < 0.05). This highlights the positive impact of FSH treatment on in vitro embryo production (IVEP) and the OPU rate.

## 1. Introduction

Assisted Reproductive Technology (ART) has revolutionized the global livestock industry, particularly in dairy and meat production. ART, namely Ovum Pick-Up (OPU) and Multiple Ovulation and Embryo Transfer (MOET) technologies, are employed extensively to enhance genetic progress in cattle breeding [1].

In the field of embryo transfer, both in vivo-derived (IVD) and in vitro-produced (IVP) embryos are utilized to achieve the same purpose through different methods. The field of embryo transfer has witnessed significant advancements, especially with regard to the production and utilization of IVP embryos acquired through OPU. The OPU technology plays a significant role in oocyte collection globally and is the most efficient means of enhancing genetic advancement in cattle through maternal lines [2,3]. According to data from the International Embryo Technology Society (IETS) newsletter report, approximately 3.47-fold more OPU-derived IVP embryos were produced than IVD embryos [4]. Since 2003, cattle embryo production has seen a remarkable increase, with a notable surge in the production of frozen IVP embryos since 2015. This increase is attributed to their improved viability and higher pregnancy rates [5]. These advancements can be largely attributed to the use of OPU. Therefore, increasing the market supply of high-quality freeze–thawed embryos, which are as stable as freeze–thawed spermatozoa, improves efficiency. Interest in the commercial utilization of OPU and in vitro embryo production (IVEP) has grown due to advancements in genomic technology, which have resulted in shortened breeding intervals and increased selection accuracy [6,7]. Therefore, in numerous studies, OPU/IVEP combined with the MOET program is actively employed to evaluate the quality of COCs acquired through OPU and to optimize the application of this technology.

Over the years, MOET has played a pivotal role in bovine embryo production, and cervical lavage collection is a highly effective method, resulting in improved survival and cryogenic resistance [7]. Follicle-stimulating hormone (FSH) activates receptors within the zona pellucida. This activation leads to an increase in the expression of aromatase cytochrome P450, which converts androgen to estrogen. Aromatase then diffuses into the granulocytes. [8]. Synthesized estrogen increases the number of FSH receptors and granulocytes via positive feedback mechanisms and induces the expression of luteinizing hormone receptors using granulocytes in collaboration with FSH [9]. Advancements in techniques and protocols for bovine ovarian hyperstimulation have improved synchronization and hormonal protocols [10,11]. While there has been progress in increasing the number of transferable embryos recovered per unit of session, there has not been a significant improvement in the average number of embryos generated per donor [10]. Moreover, although the efficiency of IVP protocols in cattle has increased, the rate of embryos with typical development is still lower compared to those generated in IVD conditions [12]. Consequently, pregnancy rates achieved using the OPU-IVP method (in vitro embryo transfer) are lower than those obtained using the IVD (in vivo embryo transfer) [1,13], emphasizing the need to develop an IVP system that can more efficiently produce embryos with a high survival rate and enhance cryopreservation and pregnancy rates. According to a previous study, oocyte retrieval using super-stimulation with FSH has been extensively used for OPU. Ovaries that undergo stimulation produce high-quality oocytes and significantly increase the number of transferable embryos [14,15,16,17]. However, studies on cryopreservation and pregnancy rates of embryos produced from oocytes obtained through OPU-IVP and superstimulation are lacking.

Therefore, in the present study, we examined the quality and yield of oocytes between the FSH group and the control group, evaluated chromatin configuration, assessed embryo productivity, checked the quality of embryos after freezing and thawing, and ultimately compared pregnancy rates after embryo transfer.

## 2. Materials and Methods

### 2.1. Animal and Management

The animal study was performed during the reproductive season (spring and fall seasons). A total of 35 Hanwoo females, with a mean age of 3.5 ± 0.3 years (mean ± standard error of the mean), were included in this study. They were divided into two groups: control (*n* = 20) and FSH (*n* = 15), all of which were in normal cycling. The Hanwoo cows, with a mean age of 36.4 ± 1.32 months and parity of 1.2 ± 0.12, had a mean BCS of 3.0 ± 0.2 on a scale of 1–5 (where 1 means very thin and 5 very fat). They were raised on a breeding farm and received a diet consisting of high-quality dry grass, balanced feed grains, water, and mineralized salt.

### 2.2. Experimental Methodology

Hanwoo cow donors were randomly divided into two groups, and the experiment was conducted six times, once every 14 days. On Day 0, during the random estrous cycle, all cows were administered a vaginal progesterone device (1.56 g progesterone; Cue-Mate, Vetoquinol, Sydney, Australia) and 2.0 mg of intramuscular estradiol benzoate (Esron, Samyang-Anipharm, Seoul, Republic of Korea). On Day 3, 5.0 mg of intramuscular PGF2α (Lutalyse, Zoetis, Brussels, Belgium) was administered. The control group received no additional treatment, while the FSH group was given intramuscular FSH (ANTRIN R10, 10AU/Ampoule, Kyoritsu Seiyaku Corporation, Tokyo, Japan) in four decreasing doses (36, 36, 24, and 24 mg) over four administrations, with a 12 h interval between each, on Days 4 and 5. On Day 7, following a “coasting” period of 36 h in the FSH group, the progesterone device was removed just before the OPU procedure, along with the control group (Figure 1).

### 2.3. Ultrasonographic Scans

Ovarian ultrasonographic scans were carried out on Day 7 using a 6.5 MHz micro-convex transducer (HM70A, Samsung Medison, Seoul, Republic of Korea). Follicles in the ovaries were classified and measured considering their diameter as small, medium, and large follicles with diameters of <5 mm, ranging from 5 to 10 mm, and >10 mm, respectively.

### 2.4. Procedures for Obtaining Oocytes under Ultrasound Guidance

Hanwoo cows were restrained in a frame, and the rectum was cleared of feces before each procedure. The external genitals were cleaned with 70% ethanol, and epidural anesthesia (2% lidocaine) was administered to prevent injury to humans or animals. Using a 6.5 MHz microconvex transducer (HM70A, Samsung Medison, Seoul, Republic of Korea) and a handle designed for high-quality transvaginal ultrasound assessment of the ovarian structure in cows. The aspiration medium during the OPU procedure consisted of 0.005 AU/mL FSH (F2293), 1 µg/mL 17β-estradiol (E4389), all from Sigma Aldrich (St. Louis, MO, USA), mixed in µL TCM-199 medium (Trusted life science partner, Toronto, Ontario, Canada). Additionally, 10% Fetal bovine serum (GIB16000-044) and 100 µM cysteamine (M6500), also from Sigma Aldrich, were included. The aspiration was performed using a 20 G disposable hypodermic needle (WTA model Agulha com Rosca injetada-20 g; Watanabe Tecnologia Aplicada, College Station, TX, USA) assembled into a vaginal handle with a stainless-steel needle guide (20 G; 0.9 × 50 mm; Terumo Europe NV, Brussels, Belgium). A vacuum pump (WTA model BV-003, Watanabe Tecnologia Aplicada, College Station, TX, USA) maintained negative pressure for aspiration, ranging between 38 and 52 mmHg. The cultures were maintained at 38.5 °C in a humidified atmosphere with 5% CO_2_.

In the subsequent step [18] the Percoll gradient technique was employed for sperm purification. Initially, spermatozoa were isolated from thawed semen straws through density gradient centrifugation using a Percoll discontinuous gradient (ranging from 45% to 90%) at 1500 rpm for 20 min. The Percoll density gradient was established by layering 1 mL of 45% Percoll solution atop 1 mL of 90% Percoll solution (Pertoft and colleagues, Seattle, WI, USA) within a 15 mL conical tube. Thawed semen was then applied over the Percoll gradient solution and the tube was subjected to centrifugation. The resultant pellet underwent two washes with capacitation Tyrode’s albumin lactate pyruvate (TALP) media base and was subsequently centrifuged for 10 min at 1500 rpm. Active motile spermatozoa from the pellet were introduced into the droplets containing mature oocytes. On Day 0, oocytes were inseminated with 1 × 10^6^ spermatozoa/mL for 18 h in an IVF-TALP medium ) covered with mineral oil (NO-100,Nidacon, Molndal, Sweden). This process took place in a humidified atmosphere with 5% CO_2_ at 38.5 °C. Following fertilization, oocytes were denuded and cultured in a two-step chemically defined culture medium. The early stage lasted for 5 days, followed by 2 days in the later-stage medium (Biocompare Inc., San Francisco, CA, USA). These cultures were maintained at 38.5 °C in an atmosphere composed of 5% O_2_, 5% CO_2_, and 90% N_2_.

### 2.5. Oocyte Characteristics: Grade and Germinal Vesicle Chromatin Status

The oocytes were graded under a microscope (M125, Lecia, Wetzlar, Germany) immediately after OPU. The morphological characteristics were rated considering four classes: I, II, III, and IV [19]. The germinal vesicle (GV) chromatin of the oocytes was observed under a fluorescence microscope after mechanical removal of the surrounding cells using 500 mL of Dulbecco’s phosphate-buffered saline (DPBS [Thermo Fisher Scientific, Waltham, MA, USA]). Following that, the oocytes were immersed in 60% methanol dissolved in DPBS and fixed at 4 °C for 30 min. Subsequently, they were stained using a 1 mg/mL solution of Hoechst 33,342 (NucBlue Live ReadyProbe Reagent, Invitrogen, Carlsbad, CA, USA). The classification of COCs took into account the integrity of the GV and the extent of chromatin compaction [19,20,21].

### 2.6. Embryo Vitrification and Warming

Blastocyst vitrification was performed on the seventh day according to previously described protocols [22,23], with minor adjustments. Embryos were handled during vitrification and warmed using a holding medium (HM) containing TCM199-HEPES + 20% fetal calf serum. The entire process was conducted in a clean room at 32 °C, with a heated surface at 39 °C. For the vitrification process, blastocysts were first immersed in a solution composed of 10% ethylene glycol + 10% dimethyl sulfoxide (DMSO), referred to as vitrification solution 1 (VS1), for a duration of 3 min. Subsequently, they were transferred to a well containing a solution comprising 20% ethylene glycol + 20% DMSO + 0.5 M sucrose, referred to as vitrification solution 2 (VS2), for a period of 45 s. The blastocysts were subsequently loaded into a cryotop device (Meyona, Sofia, Bulgaria) along with 0.2 µL of VS2 and immediately immersed in LN_2_ for preservation. To initiate the warming process, the pulled end of the straw was directly placed into 1.2 mL of 0.25 M sucrose in HM. After a 5 min interval, the blastocysts were transferred to a 0.15 M sucrose medium in HM for an additional 5 min and then subjected to two washes with HM solution. Following the warming process, the blastocysts were rinsed with the later-stage culture medium and positioned in wells containing the same medium. At the 24 and 48 h marks during the culture, the survival rates of the warmed embryos were evaluated based on morphological criteria, encompassing the integrity of the embryo membrane, zona pellucida, and re-expansion. Additionally, we documented the percentage of embryos that resumed development and progressed to an advanced stage during culture, and their hatching rates.

### 2.7. Cell Counts before Vitrification and after Warming

At different stages, embryonic cells from the blastocysts were counted, including fresh blastocysts on days 7 and 8 after vitrification and warming, and on the day of re-expansion or hatching during IVC. Blastocysts with diameters >150 µm were selected for total cell count evaluation. The blastocysts were subjected to Hoechst 33,342 dye exposure (NucBlue Live ReadyProbe Reagent; Invitrogen, Carlsbad, CA, USA) at a concentration of 1 µg/mL for 5 min, transferred to a slide, and covered with a coverslip. Using a fluorescence stereomicroscope (Stellaris, Leica, Wetzlar, Germany) with excitation and emission wavelengths of 358 and 461 nm, respectively, the slides were examined. Microscope imaging software (LAS X, Leica, Wetzlar, Germany) was employed for the enumeration of cell nuclei.

### 2.8. Embryo Transfer and Pregnancy Assessment

On days 7 and 8, embryos were categorized based on standard, stage, and grade [24]. Embryos classified as IETS stages 5–8 and grade 1, whether fresh or frozen/thawed, were transferred, taking into account both the quantity of transferable embryos on day 7 and the number of available recipients. A single bovine embryo was loaded onto a sterile 133-mm straw containing serum-free transport blastocyst medium (Transport VitroBlast, ART Lab Solutions, Adelaide, Australia) in two microdrops. To transport the loaded straw, we utilized an embryo transporter (TE 100 Compact, Watanabe Tecnologia Aplicada, College Station, TX, USA) to the Gyeongsangbuk-do Livestock Research Institute. The transcervical method was employed for the nonsurgical transfer of a loaded straw into the uterine horn of the recipient on the 7th day of the estrous cycle [25]. Pregnancy and embryo survival rates were assessed through rectal palpation and ultrasonography on day 50 and subsequently at regular intervals during pregnancy.

### 2.9. Statistical Analysis

The present study analyzed factors such as oocyte recovery, follicle proportion, oocyte quality, chromatin configuration, embryo production, and pregnancy rates. The data were analyzed using a one-way ANOVA with a general linear model. The viability rate and number of blastocysts before and after vitrification were assessed using a Chi-square test. The results are presented as mean ± standard deviation. Statistical significance was defined at the level of *p* < 0.05.

## 3. Results

### 3.1. Ovarian Response to Hormonal Treatment

One hundred and forty-two and 90 OPU sessions were performed in the control and FSH groups. The experimental results clearly indicated that the treatment had a significant impact (*p* < 0.005) on the number of oocytes retrieved in the FSH group (12.7 ± 1.3) compared to the control group (8.5 ± 1.2) (Figure 2A). Furthermore, the FSH group exhibited a higher proportion of medium-sized follicles (6–10 mm) at 51.3%, which significantly increased their developmental potential and recovery compared to the control group (*p* < 0.0001) (Figure 2B).

Comparing the grades of the COCs in the recovered oocytes between the FSH (1125 oocytes) and control (1022 oocytes) groups, we found that the proportion of class I and II oocytes (88.2%) significantly increased in the FSH treatment group. In contrast, the percentage of class III and IV oocytes (7.2%) exhibited a significant reduction compared to the control group. Notably, the FSH group displayed a markedly higher percentage of high-grade oocytes than the control group (*p* < 0.05) (Table 1).

### 3.2. Collecting Oocytes and Producing Embryos In Vitro

After the OPU sessions, oocytes were evaluated using a stereomicroscope for selection. Oocytes were selected for subsequent in vitro experiments considering their GV chromatin status. Oocytes without cumulus cells and those with visible cytoplasmic degeneration were excluded from further analysis.

The control group yielded 80 oocytes (8.5 oocytes/cow), with the majority at the GV1 (30.1%) and GV2 (34.1%) stages and few at the GV3 stage (12.6%). Contrastingly, treatment significantly (*p* < 0.05) increased the oocytes yield in the FSH group, yielding 58 oocytes (12.7 oocytes/cow), with most at the GV2 (64.0%) stage and a few at the GV3 (20.8%), GV1 (4.4%), and GV breakdown (3.3%) stages. Furthermore, the FSH group yielded no GV0 oocytes (Figure 3). The present study revealed significant differences in the GV chromatin status and embryo quality between the FSH and control groups. Oocytes from both groups exhibited classifications into classes I, II, and III. However, the FSH group displayed higher rates of embryos reaching the 2- and 8-cell cleavage stages, as well as a notably elevated blastocyst rate compared to the control group (*p* < 0.05) (Table 2). These findings suggest that FSH stimulation led to enhanced ovarian response and improved oocyte collection within the FSH group compared to the control group.

### 3.3. Assessment of the Survival Rates of Blastocysts after Vitrification and Warming and Analysis of Staining

Embryo grades were determined on day 7 based on the criteria in the IETS manual. After vitrification and warming of the IVC for 48–72 h, the embryos produced by the FSH group re-expanded at higher rates (grade 1: 86.9%; grades 2 and 3: 57.9%) than those produced by the control group (grade 1: 75.6%; grades 2 and 3: 38.9% [*p* < 0.05]). The hatching rates were higher in the FSH group (grade 1: 76.2%; grades 2 and 3: 53.8%) than those in the control group (grade 1: 58.9%; grades 2 and 3: 26.7% [*p* < 0.05]) (Table 3).

After warming, all thawed embryos were evaluated during IVC and those with diameters >170 µm were subjected to total cell count analysis. The total cell count before vitrification was higher in the FSH group (before: 193.2 ± 43.6) than that in the control group (before: 160.1 ± 28.2). However, after warming, the total cell count decreased in both groups, with the FSH group (after: 167.1 ± 39.1) having a higher count than the control group (after: 131.4 ± 26.7). There was a notable distinction between the two groups when the total cell count after warming was considered (*p* < 0.05) (Table 4).

### 3.4. Comparison of Pregnancy Rates after Embryo Transfer to Hanwoo Cows

We transferred 567 fresh and 257 post-thaw embryos (stages 6–8 and grade 1) into 824 Hanwoo cows. Pregnancy was assessed through rectal palpation 50 days after embryo transfer, and the pregnancy rates were subsequently compared between the control and FSH groups. The control group achieved a pregnancy rate of 45.1% (186/412) and 36.2% (37/102) for fresh and post-thawed blastocysts, whereas those of the FSH group were 60.6% (94/155) and 50.9% (79/155), respectively (Table 5). The FSH group significantly improved (*p* < 0.05) in pregnancy rates for fresh and frozen/thawed embryos, with increments of 15.5 and 14.7%, respectively.

## 4. Discussion

In this study, we assessed the follicular population, oocyte recovery rate, and embryo productivity of Hanwoo donor cows that underwent FSH priming before OPU. Additionally, we distinguished between fresh and post-thaw embryos to assess the final pregnancy rate. FSH treatment during OPU resulted in an increase in the proportion of intermediate- and large-sized follicles in all categories of Hanwoo donor cows. FSH treatment resulted in a higher proportion of viable oocytes suitable for cultivation and an improved ratio of embryos produced per OPU/IVEP session. Furthermore, FSH treatment enhanced post-thaw survival, leading to increased recipient pregnancy rates. These findings support the hypothesis that FSH treatment enhances IVEP and pregnancy rates in various Hanwoo donor cows using OPU/IVEP.

Currently, commercial-scale IVP in Korea and Brazil predominantly involves OPU performed on randomly selected days of the estrous cycle, during which all visible follicles are aspirated [26,27,28,29,30]. Distinguishing between developing and atretic follicles during this process is practically impossible, and follicles of different sizes can be aspirated without bias. Consequently, a portion of atretic follicles may unavoidably be aspirated, resulting in a decrease in oocyte quality and the efficiency of embryo production [31]. The advantages of wave synchronization are readily apparent, as they accurately determine the optimal time to aspirate growing follicles.

The stage of the follicular wave at the time of OPU significantly influences the number and quality of collected COCs [32]. In this study, hormonal treatment was administered to the donors prior to the OPU procedure, resulting in a 4.2-fold increase in the number of collected COCs compared to the control group. These findings provide further support for the idea that the increased follicular diameter due to FSH treatment enhances oocyte quality and IVEP efficiency in Hanwoo donor cows, emphasizing the practical applicability of the findings from the present study. However, previous studies in *Bos taurus* and *Bos indicus* have indicated that FSH treatment raises the percentage of intermediate- and large-sized follicles, but not oocyte recovery rates [33,34,35,36,37]. Hanwoo cows, although classified as *Bos taurus*, exhibit distinct follicular wave characteristics, with a predominance of small follicles during their regular random estrous cycle. We attributed the increased oocyte recovery rate to the ease of aspirating enlarged follicles, which are at a developing stage, owing to the unique characteristics of Hanwoo cows. Furthermore, it is advisable to increase both needle diameter and pressure when conducting OPU procedures in FSH-treated donors. This adjustment helps with the aspiration of enlarged follicles. [36]. These results indicate that increased follicular diameter due to FSH treatment enhances oocyte recovery rates and IVEP efficiency [32].

The present study demonstrated the efficacy of hormonal protocols for wave synchronization in Hanwoo cows in increasing embryo numbers and proportions. This was attributed to improved control of follicular recruitment and growth stages in the ovaries. Consequently, donor FSH treatment has been proposed to favor the recovery of competent oocytes, leading to increased embryo development rates, improved productivity, and a higher number of usable oocytes. The present study demonstrated the efficacy of hormonal protocols for wave synchronization in Hanwoo cows by increasing embryo numbers and proportions, which was attributed to the improved control of follicular recruitment and growth stages in the ovaries. Consequently, donor FSH treatment has been proposed to favor the recovery of competent oocytes, leading to increased embryo development rates, greater productivity, and a higher number of usable oocytes. The observed enhancement in embryo productivity and recovery rates further supports the notion that FSH treatment positively affects oocyte quality, thus contributing to improved embryonic development. The increased embryo yield per OPU session in the FSH group highlights the potential of FSH treatment to optimize IVEP. This finding holds significant implications for assisted ART and commercial IVP programs.

The ultimate measure of success in ART is the establishment of a successful pregnancy. In this study, we assessed the pregnancy rate by transferring the produced embryos into recipient cows. In the fresh blastocyst, the FSH group exhibited a higher pregnancy rate (60.6%) compared to the control group (45.1%), and in the post-thaw blastocyst; although the overall pregnancy rate decreased compared to that observed with fresh blastocysts, the FSH group still showed a significantly higher pregnancy rate (50.9%) than the control group (36.2% [*p* < 0.05]). The higher pregnancy rate in the FSH group could be attributed to increased oocyte yield, improved oocyte quality, and enhanced embryo productivity. The enhanced survival of frozen/thawed embryos due to FSH treatment may have played a role in the elevated pregnancy rate observed in the FSH group. Using FSH-treated embryos in the vitrification process contributed to greater stability in survival rates and increased the chances of a successful pregnancy following embryo transfer. In contrast to studies involving *Bos indicus* breeds, the FSH treatment group exhibited a higher pregnancy rate compared to the control group in this study. For example, Nelore cows exhibited a pregnancy rate ranging from 33 to 36% for IVF embryos and a reported pregnancy rate of 27.2 to 31.4% [2,38,39]. In contrast, our study on Hanwoo cows (*Bos indicus coreanae*) exhibited similar advantages in embryo observation, and the FSH group exhibited a 15% higher average number of pregnancies compared to the control group. Moreover, embryos produced from FSH-treated donors had stable pregnancy rates when evaluated based on fresh embryo status. Although these findings are in contrast with those reported in studies on other breeds, they can be attributed to the considerable effects of follicular wave synchronization and superovulation in Hanwoo cows, enabling high-quality oocytes and embryos with easier in vitro maturation. However, further research is required to confirm these results. Slow freezing and vitrification are not considered safe for embryos. Slow freezing involves the transition of the fluid lipid part of the cell membrane into a gel topology, termed lipid phase transition [40], while vitrification involves the rapid cooling of embryos in liquid nitrogen using a high concentration of cryoprotective agents. Vitrification can cause severe damage to the lipid content within embryo cells and has been reported to have limited lipid content within the inner cell mass (ICM) [41]. Despite significant advances in the techniques used for developing and conserving bovine embryos, further research is required to improve these methods and enhance embryo growth and freezing efficiency [42,43]. In this study, we observed that embryos from the FSH-treated group exhibited an 11.3% increase in re-expansion and a significant 17.3% increase in hatching rate compared to the control group after freeze–thaw treatment (Table 3, *p* < 0.05). This indicates that embryos produced with FSH treatment before oocyte retrieval showed increased post-thaw survival rates. Furthermore, to support these results, we assessed the total cell count of embryos before and after vitrification. Although both groups showed a decrease in cell count after vitrification, the FSH-treated group exhibited a noticeable difference in total cell count before and after vitrification compared to the control group (Table 4, *p* < 0.05). Consequently, the increased production of embryos after FSH treatment, which led to a higher yield of viable oocytes, resulted in improved freeze–thaw survival rates in the FSH-treated group. [44,45].

In *Bos indicus* cattle, using OPU, an average of 18–25 COCs were recovered per head [38,46]. On average, *Bos indicus* has more follicular waves and follicles larger than 5 mm compared to *Bos taurus*. Furthermore, *Bos indicus* exhibits a higher COC recovery rate than *Bos taurus* [32]. This explains the observed lower average COCs recovery rate in this study, as it focused on Hanwoo cattle (*Bos taurus coreanae*). Efficient production of transplantable embryos might be attributed to FSH treatment and ovulation synchronization. However, this indicates that there has been extensive research on FSH dosage, coasting periods, repetition frequency, and grouping, and that these topics will continue to be thoroughly discussed. When compared to IETS statistics, in this study, recovered oocytes (12.7) with FSH treatment are approximately 5.7 oocytes lower than the worldwide average of recovered oocytes (18.4) [4]. However, we recorded a higher number of transplantable embryos per head (5.4) than the global average (4.4). Our research highlights the appropriate use of FSH treatment to enhance the productivity of transplantable oocytes and improve pregnancy rates in OPU, emphasizing its role in facilitating oocyte collection and subsequent embryo production. Furthermore, our findings demonstrate the potential of OPU and MOET technology to enhance genetic progress in livestock breeding through efficient reproduction and embryo transfer procedures. Ultimately, from the perspective of operators applying OPU technology, the qualitative and quantitative growth of recoverable oocytes and transplantable embryos per session is expected to result in time, labor, and cost-saving effects.

## 5. Conclusions

In the present study, FSH treatment reduced working hours compared with those in previous studies. Additionally, superovulation with FSH positively affected COC acquisition, ovulation size, and embryo production, leading to greater-than-expected values in post. Therefore, the direct effects of FSH treatment were revealed. Our findings indicate that FSH enhances embryo production, cryopreservation, and maintenance throughout the OPU-IVEP procedure.

## Figures and Tables

**Figure 1 vetsci-10-00578-f001:**
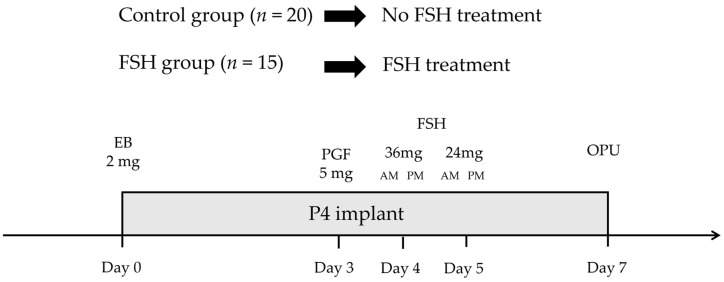
Experimental diagram. Hanwoo cow donors (Control group = 20, FSH group = 15) were randomly divided into two groups, and the experiment was conducted six times. Both of these groups underwent synchronization. Subsequently, the control group received no additional treatment, while the FSH group received 200 mg of intramuscular FSH injections divided into four doses, administered at 12 h intervals on Days 4 and 5. EB = estradiol benzoate; 1.0 mg, PGF = D-tromethamine; 5.0 mg, P4 = progesterone; 1.56 g, and FSH = follicle-stimulating hormone; the doses were 36, 36, 24, and 24 mg.

**Figure 2 vetsci-10-00578-f002:**
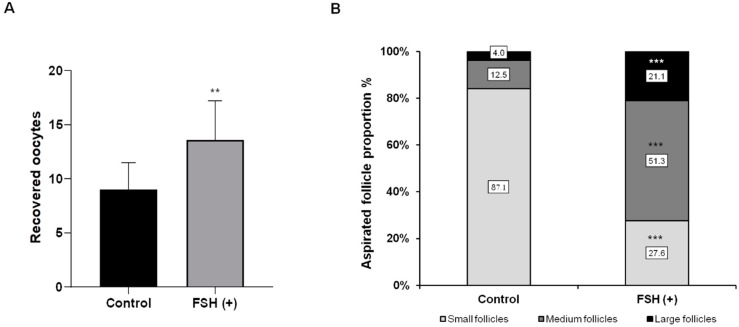
Numbers of oocytes aspirated using Ovum pik-up (OPU) (**A**) and proportion of small (10 mm) follicles (**B**) in Hanwoo donors subjected to OPU-in vitro embryo production with or without previous follicle-stimulating hormone super-stimulation ** *p* < 0.005, *** *p* < 0.0001, relative to control.

**Figure 3 vetsci-10-00578-f003:**
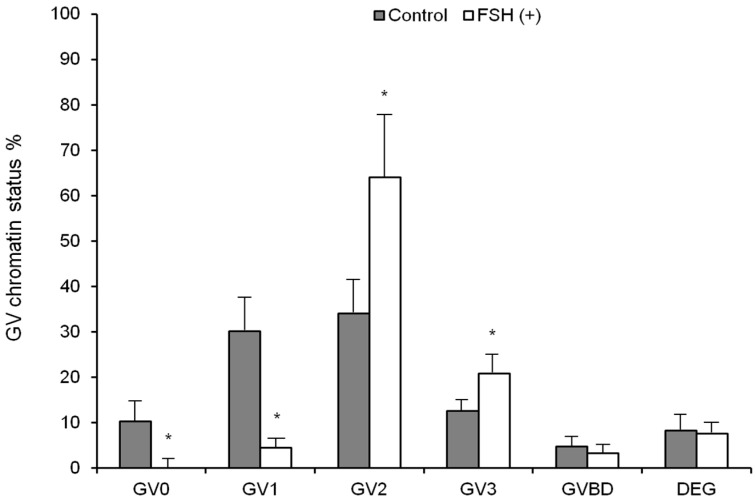
FSH treatment and the aspiration of all follicles ≥ 2 mm on a random day (day 0) and the impact of FSH treatment on day 2 regarding the patterns of chromatin configuration in oocytes obtained via OPU, specifically focusing on GV0–3 stages with increasing chromatin compaction. GV (germinal vesicle), GVBD (germinal vesicle breakdown), and DEG (degenerated oocytes). Values are the mean ± SD of per group. * = *p* < 0.05 versus Control group.

**Table 1 vetsci-10-00578-t001:** Quality of recovered oocytes from the control and FSH groups, expressed as the number and percentage of oocytes classified in each category.

	No. of Oocytes (Session)	Grade I	Grade II	Oocyte Grade (I + II)%	Grade III	Grade IV	Oocyte Grade (III + IV)%
Control	1022 (120)	321 (31.4)	193 (18.8)	514 (50.2)	330 (32.3)	178 (17.5)	508 (49.8)
FSH (+)	1125 (90)	702 (62.4)	342 (30.4)	1044 (88.2)	72 (6.4)	9 (0.8)	40 (7.2)

**Table 2 vetsci-10-00578-t002:** Summary of OPU and fertilization, and developmental capacity of the oocytes collected in the control and FSH groups.

Group	Number of Oocytes Generated by IVM (Session)	Number of Developed		
		Cleaved Embryos (%)	8-Cell Stage (%)	Blastocysts (%)
Control	884 (120)	621 (70.2)	464 (52.4)	275 (31.1) ‡
FSH (+)	1116 (90)	917 (82.2)	828 (74.2)	499 (44.7) †

†,‡ Different symbols within the same column represent significant differences based on the Chi-square test (*p* < 0.05).

**Table 3 vetsci-10-00578-t003:** Assessment of re-expansion and hatching rates for vitrified and warmed blastocysts from both the control and FSH groups.

Quality Grade	Group	Number of Cultured Embryos	Number (%) of Developing Embryos		
			in Culture	Re-Expansion	to Hatched Blastocyst
1	Control	90	82 (91.1)	68 (75.6)	53 (58.9) ‡
	FSH (+)	84	82 (97.6)	73 (86.9)	64 (76.2) †
2–3	Control	90	68 (75.6)	35 (38.9)	24 (26.7) †
	FSH (+)	52	43 (82.7)	33 (57.9)	28 (53.8) †

†,‡ Different symbols within the same column represent significant differences based on the Chi-square test (*p* < 0.05).

**Table 4 vetsci-10-00578-t004:** Total cell number of blastocysts in the control and FSH groups before and after vitrification and warming.

Group	Blastocysts Diameter µm (±SD)	Time	No. of Blastocysts	Total Cell Number Mean (±SD)
Control	186 (±34.8) †	Before vitrification	54	160.1 (±28.2) ‡
		After warming	46 §	131.4 (±26.7) †
FSH (+)	194 (±32.2) †	Before vitrification	69	193.2 (±43.6) †
		After warming	63 §	167.1 (±39.1) †

†,‡ Different symbols within the same column represent significant differences based on the Chi-square test (*p* < 0.05). § Represents embryos that were evaluated for total cell numbers, accounting for potential losses during staining or difficulties in observation. SD—standard deviation.

**Table 5 vetsci-10-00578-t005:** Comparison of pregnancy rates embryos transfer to Hanwoo cows in control and FSH groups.

Group	No. of Farms	Blastocysts	Number of Recipients Transferred	Number of Recipients Pregnant (%)
Control	45	Fresh blastocysts	412	186 (45.1)
		Post-thaw blastocysts	102	37 (36.2)
FSH (+)	31	Fresh blastocysts	155	94 (60.6)
		Post-thaw blastocysts	155	79 (50.9)

## Data Availability

The data presented in this study are available upon request from the corresponding author.

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
