# Peer review of "Improving Cryopreservation Efficiency and Pregnancy Rate through Superovulation with Follicle-Stimulating Hormone in Korean Hanwoo Cows via Ovum Pick Up"

_vetsci, 2023, doi:10.3390/vetsci10090578_

Round 1
Reviewer 1 Report
The study is well-written and easy to understand.
It was found that the superovulation with FSH improved the oocyte quality they collected after the OPU and positively affected COC acquisition, ovulation size, and embryo production.
Small improvements and additions are necessary. Otherwise, the study can be published.
The below minor improvements are necessary:
​ Table 2: The P-value shall be written below
Table 5 should also compare fresh blastocysts with fresh blastocysts and post-thaw blastocysts with post-thaw blastocysts (control versus FSH) and discuss the results.
The discussion does not come out well. Why are these results necessary for practice? Where to apply the results. If the authors explain in a few sentences the importance of the publication would be better presented.
Reviewer 2 Report
The manuscript by Daehyun Kim and Junkoo Yi describes the improvement of cryopreservation efficiency and pregnancy rate through superovulation with follicle-stimulating hormone in Korean Hanwoo cows via ovum pick-up. The research is innovative and interesting however, the study has some points that need to be clarified.
2.- Materials and Methods
Concerning the experimental methodology:
In Figure 1 the authors specify the pharmacological treatment and, indirectly, the time between each ovum pick-up. However, the authors only point out that the animals in the control group did not receive pharmacological treatment and that the ovum pick-ups were performed randomly during the estrous cycle. It would be interesting for the authors to report the frequency of pick-ups in each animal. Also, if no treatment was given, at what time of the estrous cycle were the pick-ups performed, or if the ovaries have a functional corpus luteum.
Also in Figure 1 "Ovun pick-up" should be written as in the rest of the text "Ovum pick-up".
Regarding the procedures for obtaining oocytes under ultrasound guidance:
Authors should specify the transducer they have used (type and frequency).
Minor editing of English language required
Round 2
Reviewer 1 Report
The authors have made corrections or additions as indicated by me. Therefore I accepted the publication.
Reviewer 2 Report
Dear Editor
Please, accept in present form.